# A New Maximum Power Point Tracking Technique for Thermoelectric Generator Modules

**Mohammed A. Qasim** [1,*] , **Naseer T. Alwan** [1,2] , **Seepana PraveenKumar** [1,*] , **Vladimir I. Velkin** [1] and **Ephraim Bonah Agyekum** [1]

1 Nuclear Power Plants and Renewable Energy Sources Department, Ural Federal University, 620002 Yekaterinburg, Russia; nassir.towfeek79@gmail.com (N.T.A.); v.i.velkin@urfu.ru (V.I.V.); agyekumephraim@yahoo.com (E.B.A.)
2 Power Mechanics Department, Kirkuk Technical College, Northern Technical University, Kirkuk 36001, Iraq
* Correspondence: mkasim@urfu.ru (M.A.Q.); ambatipraveen859@gmail.com (S.P.)

**Abstract:** Thermoelectric generators (TEGs) are devices that convert heat into electricity. This paper is about the design of a Maximum Power Point Tracking (MPPT) technique for a TEG module. The module is built using 204 TEGs connected in series. It is connected to the load through a DC/DC boost converter. The MPPT technique used in this work is the Interval Type 2 Fuzzy Logic Controller (IT2FLC). To verify its performance, the IT2FLC is compared with a traditional Perturb and Observe (P&O) MPPT algorithm in the case of power and voltage response at steady state, load switching, and through various ranges of temperature differences ($\Delta T$). The TEG module is modeled and the whole system is simulated successfully using MATLAB SIMULINK R2017a.

**Keywords:** boost; DC/DC; Interval Type 2 Fuzzy Logic System; MPPT; thermoelectric generator; TEG module

## 1. Introduction

The thermoelectric generator (TEG) was first introduced by the scientist, Thomas J. Seebeck, in 1821 [1] when he originally discussed the Seebeck effect. This phenomenon is observed with the application of heat to two types of semiconductors. During heating of the semiconductors, energy flows from the high-temperature side to the low-temperature region with a simultaneous flow of electrical current [2]. Nowadays, TEGs are manufactured in small pieces and coated with ceramic material on two sides to promote heat conduction. TEGs are also known as Seebeck generators [3]. In another approach, if a potential difference is applied to a TEG, then it will convert this potential difference into heat appearing at its two sides with different heat surfaces. In this case, the manufactured small pieces are called Peltiers [4]. The current paper examines TEG's capability to convert various temperature differences applied to its surfaces to a voltage. Numerous researchers have worked on methods to extract the maximum power from a TEG through the utilization of Maximum Power Point Tracking (MPPT) algorithms. K.N. Khamil et al. [5] did analytical modeling and simulation of Peltiers and Seebeck generator modules. The analysis was done in terms of the main parameters needed for quick evaluation. These included current, voltage, thermal resistivity, and efficiency. B. Bijukumar et al. [6] developed a controller to modulate the current of a grid-connected inverter, where the supply source is an array of TEGs. The design involved a TEG array, boost converter, and three-phase voltage source converter connected to a grid. A. Belkaid et al. [7] modeled a TEG with an MPPT technique. The MPPT uses a sliding mode control to a boost converter. This technique is compared to a Perturb and Observe (P&O) MPPT algorithm. The results showed that the efficiency of this technique is more than 2% higher than the P&O algorithm. N. Kanagaraj et al. [8] developed a variable fractional order, fuzzy logic controller based on an MPPT technique in which the operating point of the TEG is moved quickly towards an optimal position

to increase the harvest of energy. The performance of their proposed MPPT technique is compared with P&O and incremental conductance (IC) MPPT algorithms. The current work considers the design of an MPPT technique for a TEG module using an Interval Type 2 Fuzzy Logic Controller (IT2FLC). The paper is arranged as follows. An introduction, modeling of a thermoelectric generator, DC/DC boost converter, and an MPPT technique for a thermoelectric generator are presented. Then, simulation results, discussion, and conclusions follow.

## 2. Modeling a Thermoelectric Generator

A thermoelectric generator (TEG) may consist of 10 to 100 thermoelectric elements that are P-type and N-type semiconductors connected thermally in parallel and electrically in series [9]. A simple diagram of a TEG is shown in Figure 1.

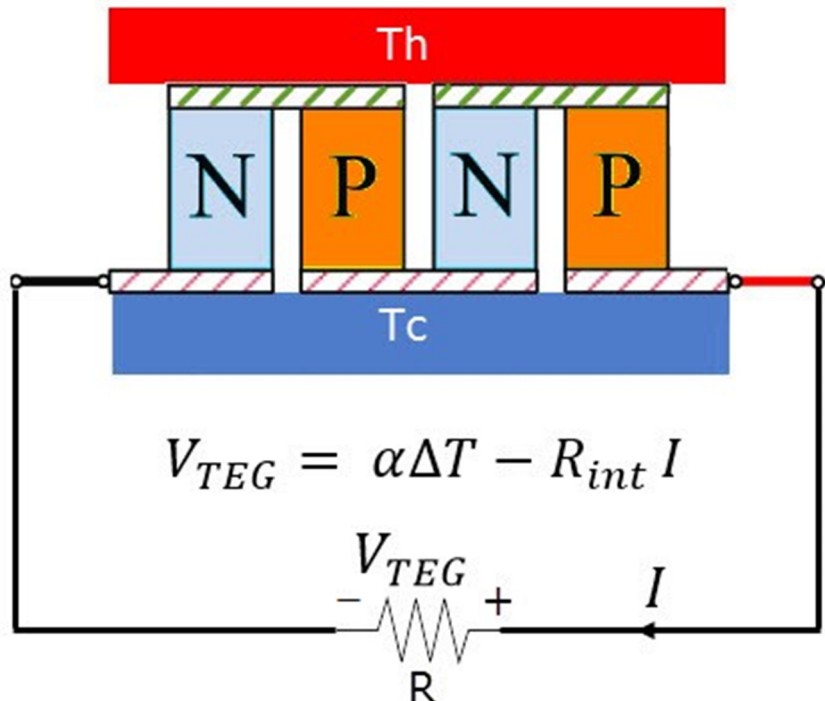

**Figure 1.** Simple diagram of a thermoelectric generator (TEG).

The operational principle of a TEG is based on the Seebeck effect. If a source of heat is applied to a TEG, a temperature difference ($\Delta T = Th - Tc$) will occur. Then, the charge carriers in the semiconductor materials, which are holes in P-type and electrons in N-Type semiconductors, will diffuse from the heat source to the cold area. As shown in Figure 1, this diffusion can cause an electric current to flow and create a voltage potential at the terminals of the TEG [10]. There are many civil and military applications of TEGs [11]. A suitable $\Delta T$ enables heat transfer to promote electron flow from the N-type to the P-type semiconductor and the generated voltage is as follows:

$$V_{oc} = \alpha \Delta T \tag{1}$$

where *Voc* is the open-circuit voltage at a TEG terminal, $\alpha$ is the Seebeck effect coefficient (V/K), and $\Delta T$ is the temperature difference between the hot *Th* and cold side *Tc* temperatures. To model the TEG in MATLAB SIMULINK, the electrical equivalent circuit of a TEG is considered. It consists of a controlled voltage source that depends on the temperature difference and Seebeck effect, connected with its internal resistance in series ($R_{int}$) [12].

A load $R$ is connected to the TEG terminal. So, the voltage equation is expressed as the following [11]:

$$V_{\text{TEG}} = \alpha \Delta T - R_{int} \, I \tag{2}$$

where $I$ is the current that flows from the TEG to the load. The maximum power transferred from TEG to the load occurs when the internal resistance of the TEG equals the load resistance. At this condition, the voltage value is known as the voltage at the maximum power point ($V_{\text{MPP}}$) where [11]:

$$V_{\text{TEG(MPP)}} = \frac{V_{oc}}{2} \tag{3}$$

To increase the voltage at terminals of a TEG module many TEGs can be connected in series or in parallel to achieve the required voltage level and power. Furthermore, at different $\Delta T$ values, the TEG module will have separate and distinct P-V curves. Figure 2 represents the P-V characteristics of a TEG module that was built based on an SP1848 TEG [13]. Moreover, for each load, there is a particular P-V curve. In the case of Figure 2, the load is 50 $\Omega$. The specifications of this module are listed in Table 1.

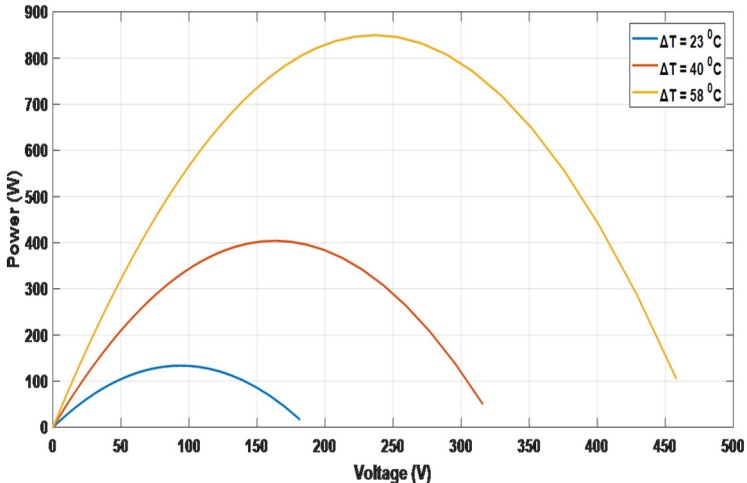

**Figure 2.** P-V curve of the proposed thermoelectric generator (SP1848) at various temperature differences.

**Table 1.** Specifications of the proposed TEG module.

| Specifications | Quantity/Value |
| --- | --- |
| No. of TEGs per Module | 204 |
| Seebeck Coefficient per Single TEG | 400 µV/K |
| Internal Resistance ($R_{int}$) of a Single TEG | 0.323 $\Omega$ |

## 3. DC/DC Boost Converter

The boost converter is a DC/DC type converter. Its output voltage is higher than its input voltage. It has an inductor (L) acting as an energy storage element, a switching device (S) that is normally IGBT or MOSFET, a diode (d) between the input and output side to direct the electricity towards the load, and input and output side filter capacitors (C1 & C2). The circuit of the boost converter is shown in Figure 3.

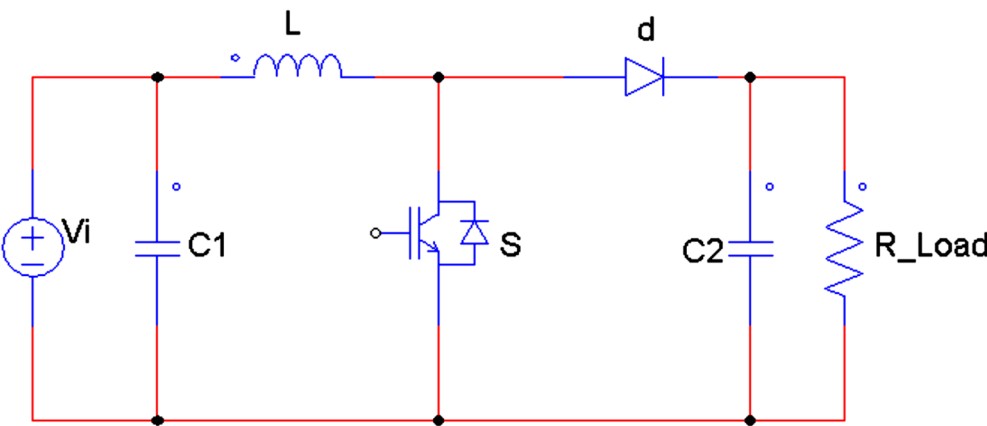

**Figure 3.** DC/DC boost converter circuit diagram.

The ON and OFF states of the IGBT switch control the amount of the output voltage [14]. When S is ON, the L will store some energy. However, if S is OFF, the inductor will discharge its stored energy towards the load to make the output voltage (*Vo*) equal the inductor L voltage plus the source voltage (*Vi*). For a loss-less system, the output voltage can be calculated as:

$$Vo = \frac{1}{1-D}Vi \tag{4}$$

where *D* is the duty cycle. The duty cycle is defined as the ratio between the period when the switch is in the ON state ($T_{ON}$) to the overall switching period ($T_{ON} + T_{OFF}$).

## 4. MPPT Technique for the Thermoelectric Generator

MPPT can be done using an algorithm, a technique, or both. As in renewable energy resources like wind and solar electricity generation systems, a TEG can use the same MPPT trends to catch its MPPs for various changes in load and temperature. One of the conventional MPPT algorithms is Perturb and Observe (P&O). This algorithm works iteratively to either increase or decrease the duty cycle of a DC/DC converter switching device. It compares the power and voltage from the previous cycle to the power of the current cycle. In the beginning of the algorithm, a starting value of power (P_pre), voltage (V_pre), duty cycle (D_pre), and the rate of duty cycle change (ΔD) should be defined. A flow chart of the P&O algorithm is shown in Figure 4 [15].

Normally in implementing the P&O MPPT algorithm, the system voltage or power may suffer from oscillations under steady-state conditions. To overcome this problem, an Interval Type 2 Fuzzy Logic Controller (IT2FLC) can be used.

The IT2FLC can work as an MPPT technique based on the P&O algorithm. Each membership function (MF) is split into two parameters, upper and lower. For MPPT purposes, there are two inputs and one output. The two inputs are the changes in power and current. The output represents the duty cycle (*D*). The MFs of the inputs are shown in Figures 5 and 6, respectively.

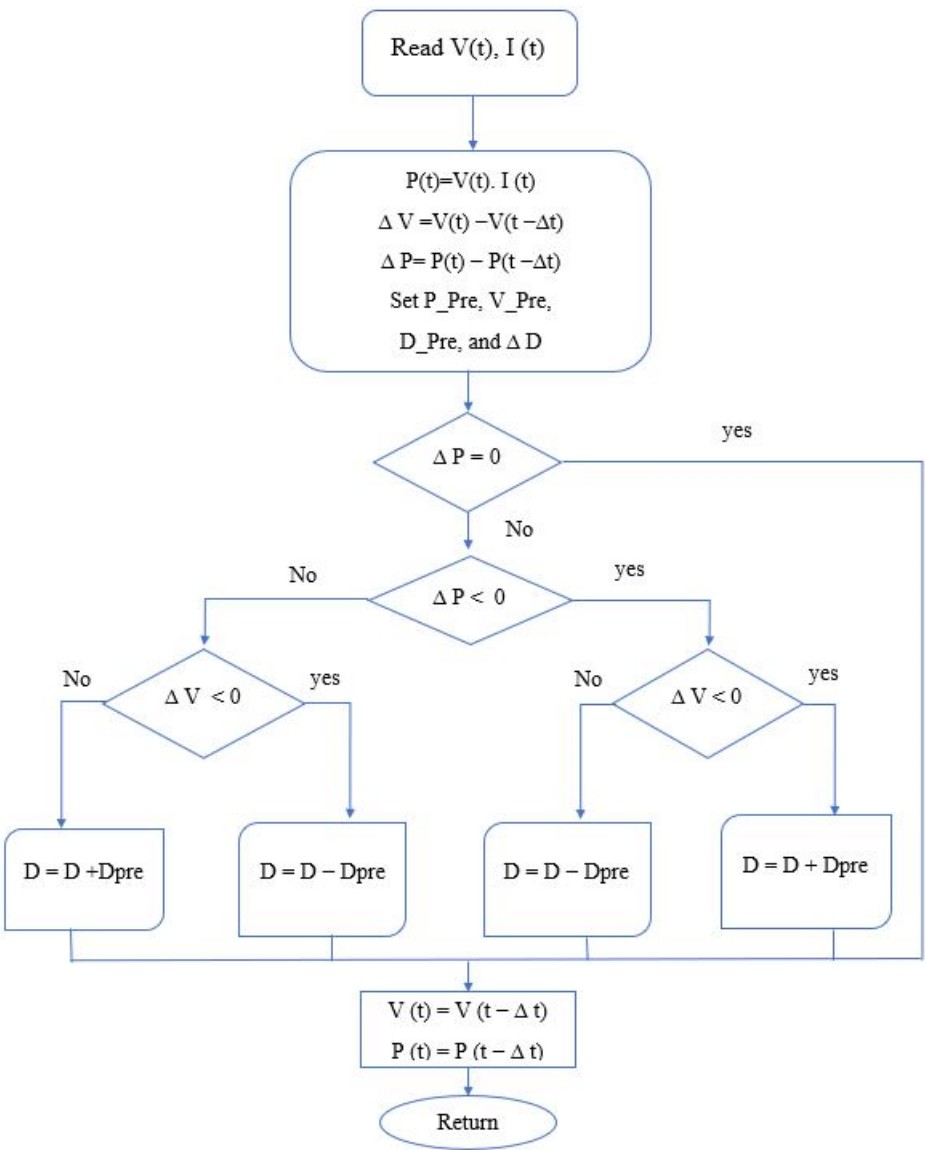

**Figure 4.** Flowchart of the Perturb and Observe (P&O) algorithm.

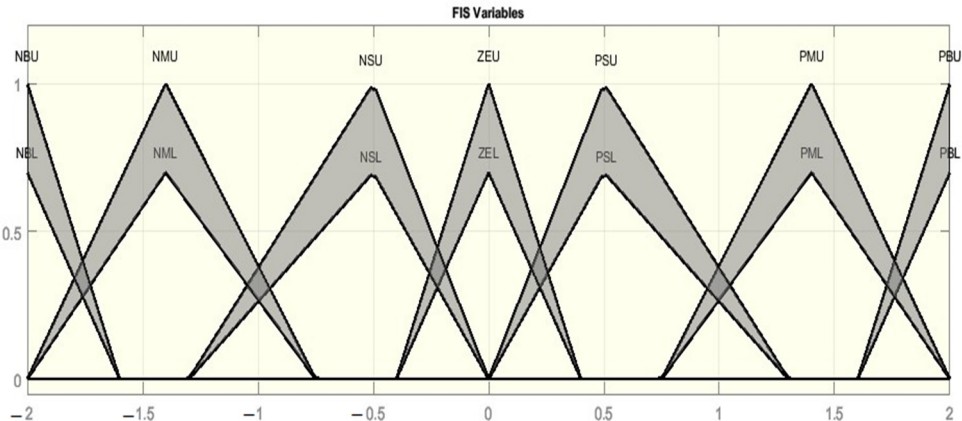

**Figure 5.** FIS variables of change in power (P).

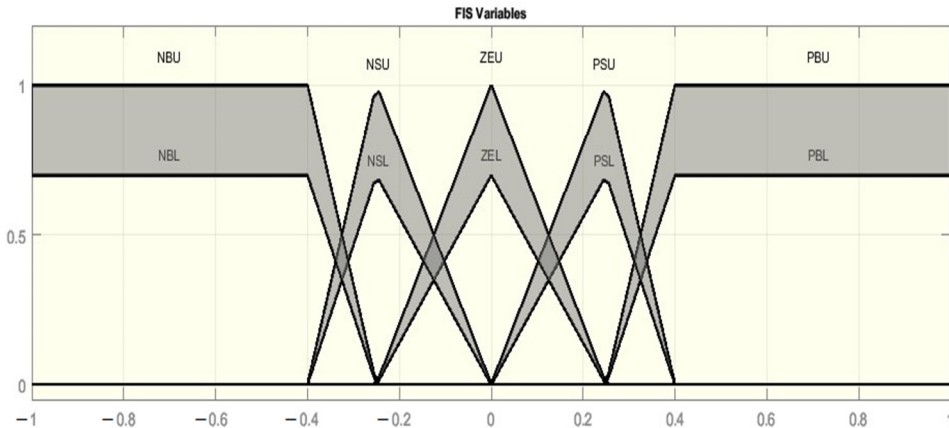

**Figure 6.** FIS variables of change in current (I).

For each input, the MFs have upper parameters represented by U and lower parameters represented as L. For the U values, the maximum degree of membership is 1, while it is 0.7 for L. There are seven MFs as inputs (change in power), and they are referenced as NB (Negative Big), NM (Negative Medium), NS (Negative Small), ZE (Zero), PS (Positive Small), PM (Positive Medium), and PB (Positive Big). All the MFs are in the range between −2 and +2. This range is known as the universe of discourse. Moreover, there are five MFs for the input (change in current), referenced as NB, NS, ZE, PS, and PB with values ranging between −1 and +1. The fuzzy interference system of IT2FLC is based on a Sugeno-type algorithm. So, the output MFs are identified in a linear vector as shown in Table 2.

**Table 2.** Output FIS variables.

| FIS Variable | Values (Linear) |
|:---:|:---:|
| NB | [0 0 -0.0075] |
| NM | [0 0 -0.003667] |
| NS | [0 0 -0.001667] |
| ZE | $[0\ 0\ -2.385 \times 10^{-19}]$ |
| PS | [0 0 0.001667] |
| PM | [0 0 0.003667] |
| PB | [0 0 0.0075] |

Since there are seven MFs for the input (change in power) and five for the input (change in current), then there are 35 rules that should be defined. These rules are given in Table 3. Defuzzification is based on the Karnik-Mendel algorithm (KM) [16]. All the MFs and rules are implemented in MATLAB through an open-source Interval Type 2 Fuzzy Logic system [17].

**Table 3.** MPPT rules for an Interval Fuzzy Type 2 system.

| | Change (P) | | | | | | |
|:---:|:---:|:---:|:---:|:---:|:---:|:---:|:---:|
| **Change (I)** | **NB** | **NM** | **NS** | **ZE** | **PS** | **PM** | **PB** |
| NB | PB | PB | PM | NM | NM | NB | NB |
| NS | PB | PM | PS | NS | NS | NM | NB |
| ZE | NB | NM | NS | ZE | PS | PM | PB |
| PS | NB | NM | NS | PS | PS | PM | PB |
| PB | NB | NB | NM | PM | PM | PB | PB |

### 5. Simulation Results and Discussion

The proposed system is simulated in MATLAB SIMULINK software version R2017a (Yekaterinburg, Russia). As shown in Figure 7, all TEGs that are connected in series are merged into one block (the Module).

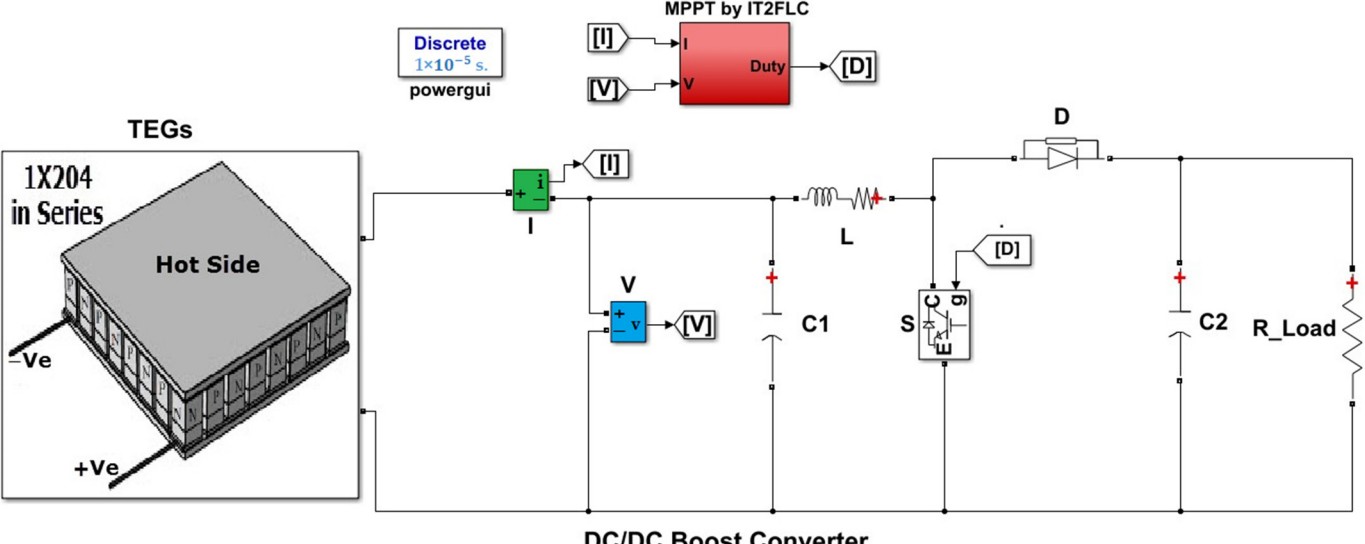

**Figure 7.** Simulink of TEG with an IT2FLC MPPT system.

In all simulations, usually, the Tc is set at 2 °C. In fact, this temperature is fixed due to flow of a cooled liquid in pipes for the low temperature side of the TEG module [18]. The TEG with a boost converter is tested when the load is 50 Ω and $\Delta T$ 48 °C, i.e., this is mean 50 °C. In the beginning, the test is made when the P&O algorithm is used, and then the IT2FLC MPPT technique is applied. In both cases, the environmental conditions and the load are the same. Figure 8 represents the input voltage and power to the boost converter when the P&O algorithm is applied, whereas Figure 9 represents the input voltage and power to the boost converter when the IT2FLC technique is employed. Comparing both figures, it can be observed that the response in each case is similar. However, when the P&O algorithm is applied, the voltage and power exhibit oscillations in their steady-state values. When the IT2FLC MPPT technique is used, the steady-state oscillation is approximately zero.

Another simulation is made to test the change in the hot side temperature or Th. For this purpose, a range of various $\Delta T$ values, 20, 30, 40, 50, 60 °C are selected. By applying the IT2FLC MPPT technique, the input power to the boost converter circuit is simulated and measured. The results are shown in Figure 10. This figure shows a fast response to a change in the hot side temperature (*Th*). As seen in Figure 10, at each change of $\Delta T$, the system reaches the new steady-state value after approximately 0.2 s and without oscillation or steady-state error.

In the current paper, a comparison is made between a manual change of the duty cycle value (*D*) of the boost converter IGBT switch (S) and when it is automatically altered in the IT2FLC MPPT technique. As shown in Table 4, a set of load values are selected, which are 100, 50, 25, 12, and 6 Ω. The $\Delta T$ is 48 °C.

In this Table, it can be noted that when the D value is increased, the power is decreased. Furthermore, when D is 10 and 20%, the power value is unchanged. The same happens when D is 30%, 40%, or 50%. The power has other values for D at 60 and 70%. When D is 80%, the power is nearly zero since, at higher D values, the conductance time of the switch is higher than for its OFF state, which results in a short circuit across the TEG terminals. Moreover, Table 4 represents the power values when the IT2FLC MPPT technique is applied using an automatic generation of D. In this Table, it can be noted that at various loads, the

power is higher than for any manual selection of D values. Furthermore, when the load is increased, the MPP is decreased.

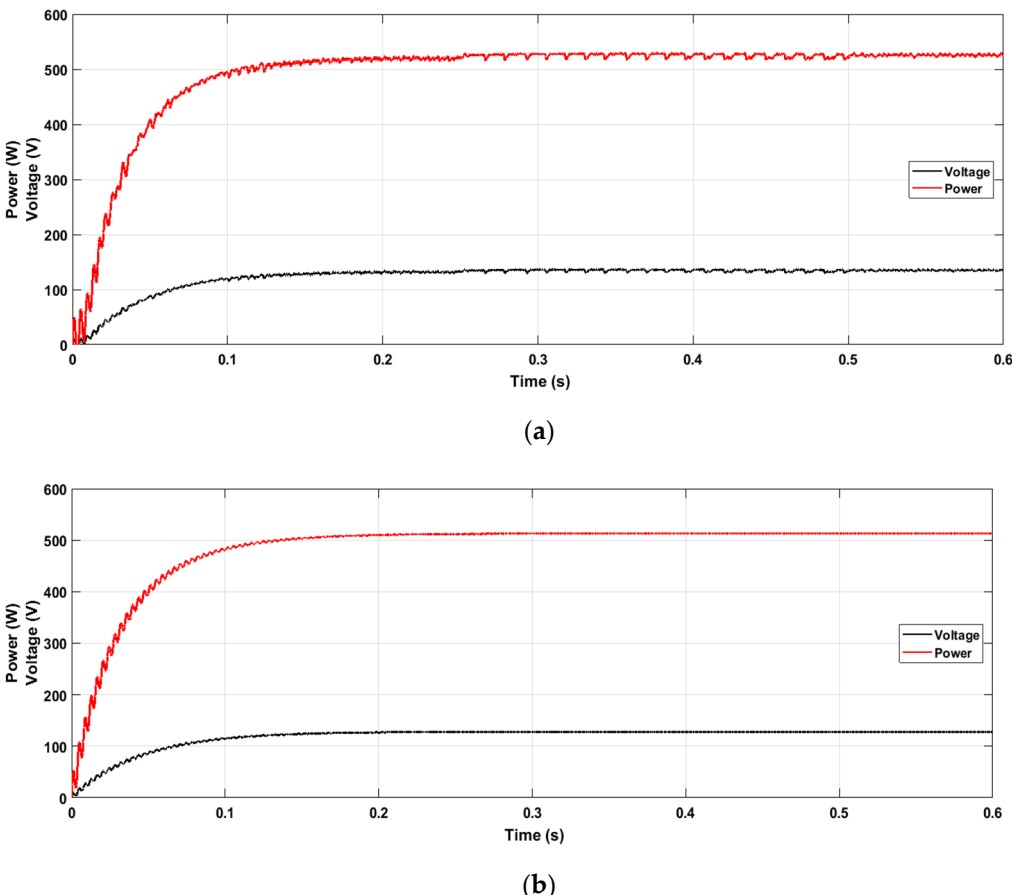

**Figure 8.** Voltage and power of the TEG with the (**a**) P&O MPPT, and (**b**) IT2FLC MPPT.

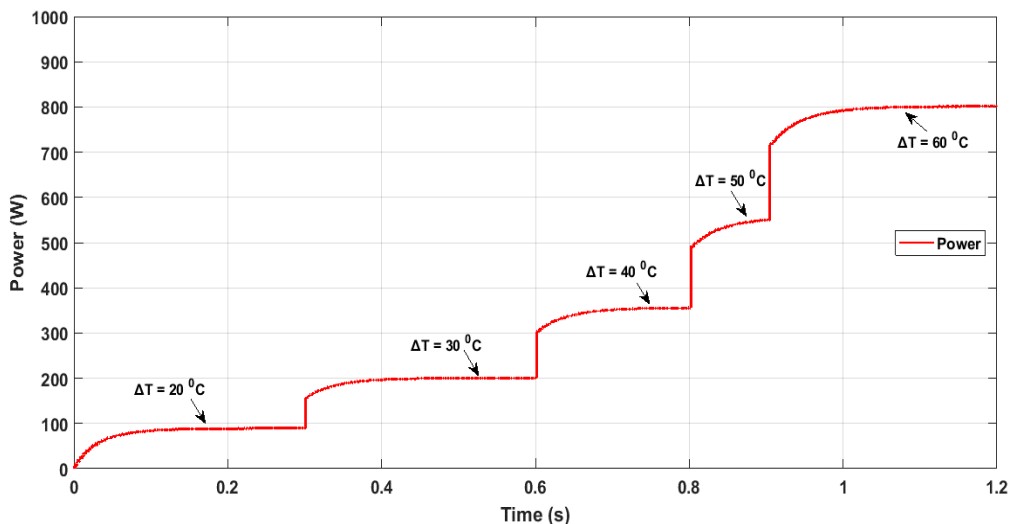

**Figure 9.** Power at various $T_h$ values when the IT2FLC technique is applied.

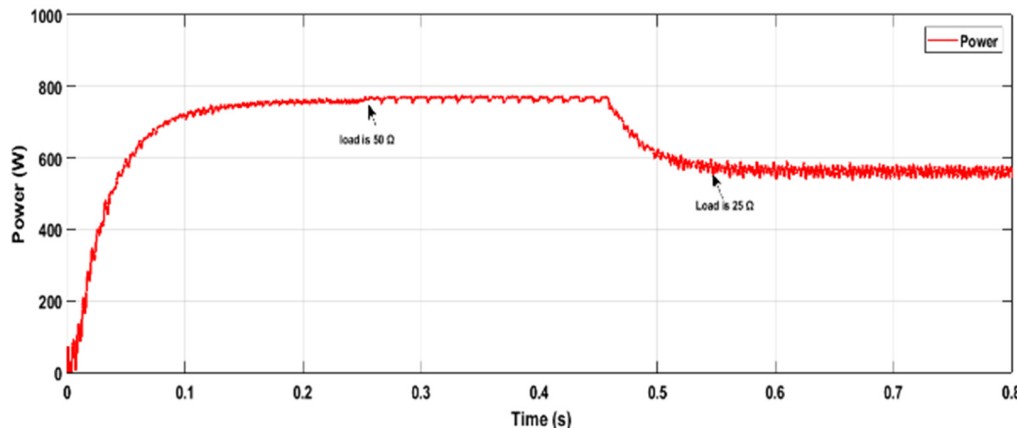

**Figure 10.** Load switching during the P&O algorithm.

**Table 4.** Manual change of duty cycle at various loads and automatic change of it at IT2FLC MPPT.

|  | Load | 100 Ω | 50 Ω | 25 Ω | 12 Ω | 6 Ω |
|---|---|---|---|---|---|---|
| D = 10% | Input Power (W) | 578.5 | 488.9 | 339.2 | 199 | 111.1 |
| D = 20% | Input Power (W) | 578.5 | 488.9 | 339.2 | 199 | 111.1 |
| D = 30% | Input Power (W) | 464.6 | 313.1 | 186.1 | 99.2 | 53 |
| D = 40% | Input Power (W) | 464.6 | 313.1 | 186.1 | 99.2 | 53 |
| D = 50% | Input Power (W) | 464.6 | 313.1 | 186.1 | 99.2 | 53 |
| D = 60% | Input Power (W) | 185.1 | 101.8 | 53.9 | 27.2 | 14.4 |
| D = 70% | Input Power (W) | 184 | 101.8 | 53.9 | 27.2 | 14.4 |
| D = 80% | Input Power (W) | 0.212 | 0.212 | 0.212 | 0.212 | 0.212 |
|  | Output Power at IT2FLC MPPT (W) | 581.4 | 511 | 368.2 | 220.5 | 124.5 |

For load switching, a comparison is made between the P&O algorithm and the IT2FLC MPPT technique. These results are shown in Figures 10 and 11, respectively.

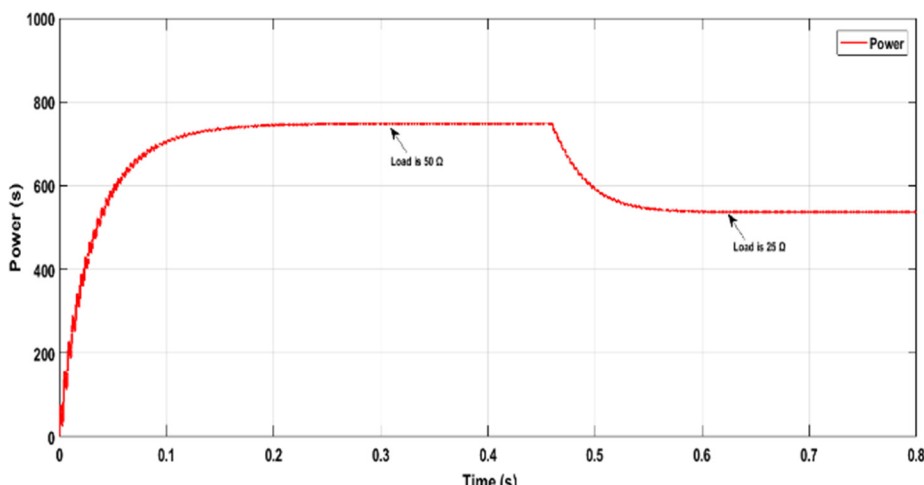

**Figure 11.** Load switching for the IT2FLC technique.

## 6. Conclusions

In this paper, a design using the IT2FLC MPPT technique is made and simulated for a TEG module. From the results, it can be concluded that the MPPT by the IT2FLC

technique has better performance than a conventional P&O MPPT algorithm in the steady-state response, changes in temperature of both sides of the TEG, and load switching. Furthermore, regarding the power values when the IT2FLC MPPT approach is used with an automated D generation, it should be noted that the power is larger at varied loads than for any manual selection of D values. Moreover, when the load increases, the MPP will also decrease. So, the IT2FLC MPPT technique can be adapted for implementation as an MPPT technique, even when merging a TEG with other renewable energy systems to form a renewable energy hybrid system. For practical implementation, it should be taken into consideration that instead of using normal water, there is the possibility of using a microfluidic heat exchanger for hot and cold surfaces for the Peltiers module. The nanofluid can be compressed via microchannel with external pumps such as Peristaltic pump, syringe pump, and pressure-driven flow controller. By using liquids that have high thermal conductivity properties of not less than 200–300 (W/m2.K-1) are also obtained. This leads to an increase in the heat flux. It is suggested in the scope of future work to use this method to increase the possibility of thermal conductivity, which thus may also lead to an increase in the generated voltage by the TEG module [18]. Furthermore, the system may need a fast processor to handle the complex calculations of the IT2FLC technique.

**Author Contributions:** Conceptualization, M.A.Q. and N.T.A.; methodology, M.A.Q.; software, M.A.Q.; validation, M.A.Q., N.T.A., E.B.A., S.P. and V.I.V.; formal analysis, M.A.Q.; investigation, M.A.Q., S.P. and N.T.A.; resources, V.I.V. and M.A.Q.; data curation, M.A.Q. and S.P.; writing—original draft preparation, M.A.Q.; writing—review and editing, M.A.Q., N.T.A.; visualization, M.A.Q., N.T.A. and S.P.; supervision, V.I.V.; project administration, M.A.Q.; funding acquisition, S.P. All authors have read and agreed to the published version of the manuscript.

**Funding:** This research received no external funding.

**Data Availability Statement:** All important data is included in the manuscript.

**Conflicts of Interest:** The authors declare no conflict of interest.

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
