# Peer review of "A New Maximum Power Point Tracking Technique for Thermoelectric Generator Modules"

_inventions, doi:10.3390/inventions6040088_

Round 1

Reviewer 1 Report

            This manuscript is written well and is a pleasure to read.  The authors should consider the following revisions:

  1. Figure 1 (really Scheme 1) should provide a visual representation of current or display the letter “I” prominently; otherwise, a label of current for “I” should appear within the circuit. A simple remedy could be to fill in the blank space in this figure with Equation 2.
  2. In Figure 2, the labels seem to be somewhat distorted by the aspect ratio used. Using the native aspect ratio should solve this problem.   
  3. The caption in Figure 4 should define the “pre” abbreviation. Similarly define in 5 and 6.  Nonetheless, the figures are positioned well with respect to the text, which defines these abbreviations.
  4. Distortions due to the aspect ratio and strangely small text appear in Figure 7.
  5. Figures 8 and 9 should be combined into Figure 8a and Figure 8b for a side-by-side comparison of the oscillatory P&O and the smooth IT2FLC.
  6. Tables 4 and 5 should be combined, with the final column the second column of Table 5.
  7. For the second sentence in Conclusions, the three aspects of better performance should be enumerated in the same order in which the corresponding figures are presented.

Author Response

First of all, we greatly appreciate your thorough and competent reviews. The reviewers have provided us with great comments and questions.  

Reviewer 1

  1. Figure 1 (really Scheme 1) should provide a visual representation of current or display the letter “I” prominently; otherwise, a label of current for “I” should appear within the circuit. A simple remedy could be to fill in the blank space in this figure with Equation 2.

Thanks for this note, the figure is corrected.

2. In Figure 2, the labels seem to be somewhat distorted by the aspect ratio used. Using the native aspect ratio should solve this problem.   

Thanks for this note; Authors believe that the labels in this figure are clear now.

3. The caption in Figure 4 should define the “pre” abbreviation. Similarly define in 5 and 6.  Nonetheless, the figures are positioned well with respect to the text, which defines these abbreviations.

Thanks for the careful reading. The caption “pre” is defined in the text before figure (4)

4. Distortions due to the aspect ratio and strangely small text appear in Figure 7.

Thanks for this note; Authors believe that the labels in this figure are clear now.

5. Figures 8 and 9 should be combined into Figure 8a and Figure 8b for a side-by-side comparison of the oscillatory P&O and the smooth IT2FLC.

Thanks for the careful reading. Both figures are combined in one figure 8a and 8b according to the reviewer suggestions

6. Tables 4 and 5 should be combined, with the final column the second column of Table 5.

Thanks for the careful reading. Both tables are combined

7. For the second sentence in Conclusions, the three aspects of better performance should be enumerated in the same order in which the corresponding figures are presented.

Thanks for this note, the order of the sentence are corrected as required as colored in “red”

Reviewer 2 Report

Qasim’s manuscript describes a design of maximum power point tracking technique (MPPT) by adding the Interval Type 2 Fuzzy Logic Controller (IT2FLC) into the traditional Perturb and Observe (P&O) MPPT algorithm. The result shows fewer oscillations at the steady-state compared to the traditional P&O method. It is well organized and written. It can be published, and the followings are a few comments.

  1. Page2, Equation 2, the alpha symbol looks odd which should be a typo, and it should be corrected.
  2. What is the reason to choose the cold side temperature of 2C, which is not mentioned in the SP1848 data sheet?

Author Response

  1. Page2, Equation 2, the alpha symbol looks odd which should be a typo, and it should be corrected.

Thanks for this note, the term  is corrected.

  1. What is the reason to choose the cold side temperature of 2C, which is not mentioned in the SP1848 data sheet?

The reason for selecting this value is based on our knowledge in other experimental work which is not mentioned in this paper. Currently, we are working on cooling the water or by use of anti-freezing liquid. It is deleted from table .1 the hint for this purpose is added after fig 7.